# Whole-Genome Resequencing Analysis of the *Camelus bactrianus* (Bactrian Camel) Genome Identifies Mutations and Genes Affecting Milk Production Traits

**DOI:** 10.3390/ijms25147836

**Published:** 2024-07-17

**Authors:** Huaibing Yao, Zhangyuan Pan, Wanpeng Ma, Zhongkai Zhao, Zhanqiang Su, Jie Yang

**Affiliations:** 1Key Laboratory of Biological Resources and Genetic Engineering, College of Life Science and Technology, Xinjiang University, Urumqi 830017, China; yaohuaibing@stu.xju.edu.cn (H.Y.); zhaozhongkai828@126.com (Z.Z.); 2Xinjiang Camel Industry Engineering Technology Research Center, Urumqi 830017, China; 3Institute of Animal Sciences, Chinese Academy of Agricultural Sciences, Beijing 100193, China; zhypan01@163.com; 4College of Veterinary Medicine, Xinjiang Agricultural University, Urumqi 830052, China; mwp086010@163.com (W.M.); szq00009@163.com (Z.S.)

**Keywords:** Bactrian camel, milk production traits, genomic analysis, SNP, association analysis

## Abstract

Milk production is an important trait that influences the economic value of camels. However, the genetic regulatory mechanisms underlying milk production in camels have not yet been elucidated. We aimed to identify candidate molecular markers that affect camel milk production. We classified Junggar Bactrian camels (9–10-year-old) as low-yield (<1.96 kg/d) or high-yield (>2.75 kg/d) based on milk production performance. Milk fat (5.16 ± 0.51 g/100 g) and milk protein (3.59 ± 0.22 g/100 g) concentrations were significantly lower in high-yielding camels than those in low-yielding camels (6.21 ± 0.59 g/100 g, and 3.93 ± 0.27 g/100 g, respectively) (*p* < 0.01). There were no apparent differences in gland tissue morphology between the low- and high-production groups. Whole-genome resequencing of 12 low- and 12 high-yield camels was performed. The results of selection mapping methods, performed using two methods (*F_ST_* and *θπ*), showed that 264 single nucleotide polymorphism sites (SNPs) overlapped between the two methods, identifying 181 genes. These genes were mainly associated with the regulation of oxytocin, estrogen, ErbB, Wnt, mTOR, PI3K-Akt, growth hormone synthesis/secretion/action, and MAPK signaling pathways. A total of 123 SNPs were selected, based on significantly associated genomic regions and important pathways for SNP genotyping, for verification in 521 additional Bactrian camels. This analysis showed that 13 SNPs were significantly associated with camel milk production yield and 18 SNPs were significantly associated with camel milk composition percentages. Most of these SNPs were located in coding regions of the genome. However, five and two important mutation sites were found in the introns of *CSN2* (β-casein) and *CSN3* (κ-casein), respectively. Among the candidate genes, *NR4A1*, *ADCY8*, *PPARG*, *CSN2*, and *CSN3* have previously been well studied in dairy livestock. These observations provide a basis for understanding the molecular mechanisms underlying milk production in camels as well as genetic markers for breeding programs aimed at improving milk production.

## 1. Introduction

The Bactrian camel, *Camelus bactrianus*, is found in cooler and drier areas of northern Asia. Since ancient times, camels have been exploited as beasts of burden and as milk, meat, and wool sources. Today, camels hold tremendous economic and medical value as a sustainable livestock species with specific features, such as immunogens and milk composition. Camels facilitate East–West trading and cultural exchange by connecting the Arabian Peninsula with the Sahara and Levant to the Far East and Asia [1]. Bactrian camel milk is highly similar to human milk, with high concentrations of insulin, unsaturated fatty acids, multivitamins (A, C, and E), and minerals and low concentrations of cholesterol and lactose [2,3]. In recent years, camel milk has been greatly favored by consumers because of its high medicinal and nutritional value.

Various Bactrian camel breeds exist in China, and they are classified based on their geographical distribution. The National Breed List of Livestock and Poultry Genetic Resources (2021 edition) divides Bactrian camel breeds into the following five Chinese landraces: Junggar, Tarim, Alashan, Sunit, and Qinghai. The main dairy camel population, the Junggar Bactrian camel, has been raised as an economic and transportation animal in Xinjiang, Northwestern China. Over time, it has evolved a few specific physiological characteristics that enable the production of beneficial items, such as milk, meat, and fur, despite the harsh environment. Camels are uniparous and have long growth cycles. Females have late sexual maturity (around three years of age) and begin mating at approximately 4–5 years of age. As slow reproducers, they give birth once every 2 years. Regarding their livestock value, the low milk production of Bactrian camels has become a major bottleneck restricting the development of the camel milk industry. Therefore, selecting and breeding camels with high milk production would facilitate adequate milk production, benefiting herders.

Traditional hybrid breeding techniques have led to limited success in improving large domestic animal production traits. Selecting livestock with economically desirable traits using molecular and genetic technology has become an essential breeding method. Identifying candidate genes and genomic regions that regulate milk production traits is essential for enhancing milk-related traits [4,5]. Casein plays a vital role in determining the characteristics of animal milk [6]. Previous studies have shown that casein genes affect milk composition during the lactation period in Kazakh Bactrian and dromedary camels [6,7,8]. There are multiple polymorphic variation sites in casein genes that can serve as effective molecular markers for camel selection programs. However, only a few studies on the genetic polymorphisms of casein genes in camel populations have been reported. A previous study showed that the phosphatidylinositol-binding clathrin assembly protein (*PICALM*) gene may be associated with milk content in dromedary camel populations [9]. Another study indicated that two single-nucleotide polymorphisms (SNPs), located in the *OSBPL8*, *MRPL37*, *SSBP3*, and *LOC102516351* genes, were significantly associated with lactose and protein content in milk from Gobi Red camels [10]. Moreover, previous transcriptome studies in our laboratory have shown that gonadotropin-releasing hormone (GnRH) secretion, phosphoinositide-3 kinase-cellular homolog of the v-Akt oncogene, S/T protein kinase (PI3K-Akt), fat digestion and absorption, MAPK, forkhead box protein o (FoxO), contraction of “Wingless” and “Int” (Wnt), and mammalian target of rapamycin (mTOR) signaling pathways have been associated with milk production traits in camels [11]. Candidate genes enriched in the above-mentioned signaling pathways could be used as potential molecular markers for milk yield and composition in camels. A previous study has shown that heritability and repeatability estimates for milk yield at 305 days were 0.24 and 0.28, respectively, in Saudi camels [12]. Selecting milk production traits with moderate heritability and gains has not been achieved with traditional breeding [13]; therefore, we propose that milk production traits are affected by multiple genes and mutations at many sites with complex interactions. An additional whole-genome scan, therefore, is warranted.

The economic traits and genetic characterization of Bactrian camels are essential prerequisites for effective conservation, utilization, and breeding programs. Milk production traits have been extensively studied in cattle, sheep, and other domestic animals but not in camels. To our knowledge, no genomic studies on the milk production traits of camels have been reported. Mammary epithelial cells can convert nutrients from the circulation into milk components. In this study, we compared the morphology of mammary tissue from high- and low-producing camels. We performed whole-genome resequencing of 24 camels which were selected to have extreme milk production performance. We also performed a sweeping analysis to identify the underlying milk production-related variants and genes in Junggar Bactrian camels. Based on the currently available whole-genome drafts, this study used selection mapping to screen for genes, SNPs, and signaling pathways associated with camel milk production traits. Our findings shed light on the molecular markers associated with camel milk production traits to link economically relevant phenotypes to genotypes and make full use of the diverse genetic resources in camels.

## 2. Results

### 2.1. Differences between High- and Low-Yield Camels

We compared the milk production traits between high- and low-yielding camels. The milk indicators are shown in Table 1 and Figure 1. At both sampling sites, the lactating she-camels were exposed to the same rearing system and environment. As expected, the high-yield group showed an average milk yield (904.83 ± 57.80 kg) that was significantly higher than that of the low-yield group (520.14 ± 59.05 kg) (*p* < 0.01) according to milk production throughout the 300-day lactation period (Appendix A). The fat and protein concentrations of milk from high-yield camels and low-yield camels were 5.16 ± 0.51 g/100 g and 3.59 ± 0.22 g/100 g (fat concentrations), and 6.21 ± 0.59 g/100 g and 3.93 ± 0.27 g/100 g (protein concentrations), respectively. There was a significant difference between the high- and low-production groups (*p* < 0.05), indicating that the chemical composition of high-yielding camel milk was lower than that of low-yielding camel milk.

The mammary glands of she-camels in lactation were fully developed with parenchyma and interstitium (Figure 2). Mammary lobules (breast acini) and udder ducts were observed in all Bactrian camel mammary tissue samples. Most mammary acinar epithelial cells were single-layered columnar or cuboidal epithelial cells. Compared to the low milk-yielding she-camels (Figure 2A), the high-yielding she-camels possessed more numerous mammary epithelial cells, which were tightly arranged with narrower cell gaps (Figure 2B). No significant differences in mammary tissue morphology were observed between the two groups.

### 2.2. Genome Sequencing, Mapping, and SNP Detection

DNA quality control was conducted, and all samples passed acceptable standards (Appendix A). Deep resequencing of 24 samples from Bactrian camels generated a total of 53.9 hundred million 150-bp paired-end reads, with an average depth of 14.4× per individual and an average coverage rate of approximately 99% (Appendix A). Data from 24 camels for whole-genome resequencing were obtained from NCBI BioProject PRJNA876601. The mapping rates of our samples to the reference genome assembly were 98.82–99.07%, indicating that the sequencing data could be used for further analysis (Appendix A). We identified 250,154,884 SNPs in all samples, with a heterozygosity range of 19–24% (Appendix A). According to the SNP results, most of the high-quality SNPs were located in intergenic (6,479,911) and intronic (3,645,806) regions, with only 49,980 synonymous and 42,993 non-synonymous SNPs located within exons (Figure 3 and Appendix A).

### 2.3. Genome-Wide Selection Mapping Analysis

In our analysis of the genomes of high-yielding and low-yielding camels, two methods (*F_ST_* and the *θπ* ratio) were used to identify SNPs and genes associated with milk production traits. The top 1% of SNPs/genes with low levels of heterozygosity and high genetic differentiation were considered candidate regions. The results were as follows: we regarded areas with an extremely low or high *θπ* ratio (1% left tail and right tail, with *θπ* ratio values of −1.089 and 1.069, respectively) and significantly high (top 1%) *F_ST_* values (0.092) as strong selective regions. Results are shown in Figure 4. We identified 264 SNPs comprising 181 genes distributed in selected overlapping regions in high milk-producing camels (Appendix A).

### 2.4. Functional Enrichment and Pathway Analysis

Based on the genes in the selected region, we identified 250 Gene Ontology (GO) terms and 207 Kyoto Encyclopedia of Genes and Genomes (KEGG) signaling pathways. The results showed that 104 GO terms were enriched for biological processes (BP), 40 GO terms were enriched for cellular components (CC), and 106 GO terms were enriched for molecular functions (MF) (Appendix A). GO enrichment analysis showed that voltage-gated cation channel activity, voltage-gated ion channel activity, voltage-gated channel activity, protein kinase activity, inward rectifier potassium channel activity, phosphotransferase activity, alcohol group as acceptor, kinase activity, extracellular region, protein phosphorylation, phosphorylation, potassium ion transport, small GTPase-mediated signal transduction, metal ion transport, defense response, intracellular signal transduction, and immune response were significantly enriched (Figure 5A). Among the top 20 KEGG pathways, genes were enriched in the oxytocin; estrogen; ErbB; Wnt; mTOR; PI3K-Akt; growth hormone synthesis, secretion, and action; and MAPK signaling pathways (Figure 5B and Appendix A). These pathways play important regulatory roles in mammary gland development, milk secretion, and expression of milk fats and proteins in animals.

### 2.5. SNP Genotyping

In the expanded experimental population (*n* = 521), we selected the top 123 SNPs in 68 candidate genes for validation (Appendix A). SNPs were classified using matrix-assisted laser desorption/ionization time-of-flight mass spectrometry (MALDI-TOF-MS). A total of 112 SNPs were successfully genotyped, and the genotype detection rate at these loci was approximately 94% based on the subsequent association analysis (Appendix A). Considering one of the SNPs as an example, there were three genotypes for the *ROR2* SNP (g.7_2434270): TT, TA, and AA. The orange, yellow, green, and blue dots represent the three distinct genotypes, whereas the red dots indicate genotyping failure (Figure 6A). Similarly, the PCR-SSCP results showed PCR banding profiles for the TT, TA, and AA genotypes (Figure 6B). We also analyzed the direct sequencing peak map of the polymerase chain reaction-single strand conformation polymorphism (PCR) products containing the target sites and found that they appeared at the mutant site (Figure 6C), which is consistent with previous findings. For *ROR2* g.7_2434270, the TT genotype was associated with higher milk production than the AA genotype, while no significant differences (*p* < 0.05) in milk production were observed between the TT and TA genotypes (Figure 6D).

### 2.6. Associations between the SNPs and Milk Production Yield

Association analysis indicated that 13 SNPs were significantly associated with milk yield in Bactrian camels (Table 2). More specifically, the milk yield of camels carrying the GG genotype of the *NR4A1* g.17_2476351 T > G polymorphism was higher than that of camels carrying the TT genotype (*p* < 0.05). Milk production yields in the experimental population with the TT genotype of the *ADCY8* g.56_11964679 C > T SNP were significantly higher than those with the CC and CT genotypes (*p* < 0.05). The milk production yields of camels carrying the TT and TA genotypes of the *ROR2* g.7_2434270 A > T SNP were higher than those of camels carrying the AA genotype, among which there was a significantly higher milk production yield from TT-genotype individuals than from AA-genotype individuals (*p* < 0.05). The milk production yields of camels carrying the CC and TT genotypes of the *NRG3* g.14_2441931_15_2441956#2 C > T SNP were higher than those carrying the TC genotype, among which there was a significantly higher milk production yield from CC-genotype individuals than from TC-genotype individuals (*p* < 0.05). The milk production yields of camels carrying the TC and CC genotypes of the *IGF1R* g.69_3052315 T > C SNP were higher than those of camels carrying the TT genotype, among which there was a significantly higher milk production yield from TC-genotype individuals than from TT-genotype individuals (*p* < 0.05). Camels with the CC and TC genotypes of the g.NW_011516269_1.40371020 C > T SNP yielded more milk than those carrying the TT genotype, among which there was a significantly higher milk production yield from TC-genotype individuals than from TT-genotype individuals (*p* < 0.05). Camels with the TT and GT genotypes of the g.NW_011516269_1.40373887 G > T SNP yielded more milk than those carrying the GG genotype, among which there was a significantly higher milk production yield from TT-genotype individuals than from GG-genotype individuals (*p* < 0.05). Camels with the TT and TC genotypes of the *RHOA* g.NW_011544898_1.7434465 C > T SNP yielded more milk than those carrying the CC genotype, among which there was a significantly higher milk production yield from TT-genotype individuals than from CC-genotype individuals (*p* < 0.05). Camels with the AA and AG genotypes of the *PCSK9* g.NW_011544898_1.7434465 A > G SNP yielded more milk than those carrying the GG genotype, among which there was a significantly higher milk production yield from AA-genotype individuals than from GG-genotype individuals (*p* < 0.05). Camels with the GT and TT genotypes of the *CRKL* g.NW_011511178_1.2476130 G > T SNP yielded more milk than those carrying the GG genotype, among which there was a significantly higher milk production yield from GT-genotype individuals than from GG-genotype individuals (*p* < 0.05). Camels with the AA and AG genotypes of the LOC105075649 g.NW_011515666_1.146942 A > G SNP yielded more milk than those carrying the GG genotype, among which there was a significantly higher milk production yield from AG-genotype individuals than from GG-genotype individuals (*p* < 0.05). The GG and CG genotypes of the g.NW_011516269_1.40374976 G > T SNP yielded more milk than those carrying the CC genotype, among which there was a significantly higher milk production yield from GG-genotype individuals than from CC-genotype individuals (*p* < 0.05). Camels with the AA and GA genotypes of the g.NW_011516269_1.40370698 A > G SNP yielded more milk than those carrying the GG genotype, among which there was a significantly higher milk production yield from AA-genotype individuals than from GG-genotype individuals (*p* < 0.05).

### 2.7. Associations between the SNPs and Milk Composition Percentages

Association analysis revealed that 18 SNPs were significantly associated with milk composition percentages in Bactrian camels (Table 3). The milk protein percentage in the experimental population with the GG genotype of the *ADCY8* g.56_11964679 G > A SNP was significantly higher than in that with the AA and GA genotypes (*p* < 0.05). The milk fat percentage in camels carrying the CC and TC genotypes of the *TMEM44* g.25_7878112_26_7878118.2#2 C > T SNP was higher than that in camels carrying the TT genotype, among which there were significantly higher milk fat percentages of CC genotype individuals than of TT genotype individuals (*p* < 0.05). The milk protein percentage in the experimental population with the GG genotype of the *DMD* g.62_3736867 A > G SNP was significantly higher than in that with the AA and AG genotypes (*p* < 0.05). The milk protein percentage in the experimental population with the GG genotype of the *PPARG* g.65_22885502 G > A SNP was significantly higher than that with the GA genotype (*p* < 0.05). The milk fat and protein percentages in the experimental population with the GG genotype of the *ROR2* g.14_2441931_15_2441956#1 G > C SNP were significantly higher than those in camels with the GC genotype (*p* < 0.05). The milk fat and protein percentages in the experimental population with the AA genotype of the *CSN2* g.2126 A > G SNP were significantly higher than those in camels with the GG and GA genotypes (*p* < 0.05). Camels with the GG and TT genotypes of the *CSN2* g.4109 G > T SNP were more common than those carrying the GT genotype, among which there were significantly higher milk fat and protein percentages in the GG-genotype individuals than in the GT-genotype individuals (*p* < 0.05). Milk fat and protein percentages in the experimental population with the GG genotype of the *CSN2* g.4321 G > T SNP were significantly higher than those in camels with the GT and TT genotypes (*p* < 0.05). The milk fat and protein percentages in the experimental population with the AA genotype of the *CSN2* g.4321 A > G SNP were significantly higher than those in camels with the AG and GG genotypes (*p* < 0.05). Milk fat and protein percentages in the experimental population with the AA genotype of the *CSN2* g.8312 A > G SNP were significantly higher than those in camels with the GG and GA genotypes (*p* < 0.05). The milk fat and protein percentages in the experimental population with the AA genotype of the *CSN3* g.1033 A > C SNP were significantly higher than those in camels with the CC and CA genotypes (*p* < 0.05). Milk fat and protein percentages in the experimental population with the GG genotype of the *CSN3* g.1229 T > G SNP were significantly higher than those in camels with the GT and TT genotypes (*p* < 0.05). The percentage of milk protein in the experimental population with the GA genotype of the *CDH7* g.NW_011509967_1-3155241 G > A SNP was significantly higher than that in camels with the AA and GG genotypes (*p* < 0.05). The GA and AA genotypes of the *CDH7* g.NW_011509967_1-3169781 A > G SNP were more prevalent than the GG genotype, with a significantly higher milk protein percentage in the GA-genotype individuals than in the GG-genotype individuals (*p* < 0.05). The AG and GG genotypes of the *PCSK9* g.NW_011514566_1.392885 A > G SNP were more prevalent than the AA genotype, with a significantly higher milk fat percentage in the GG-genotype individuals than in the AA-genotype individuals (*p* < 0.05). The CC and CT genotypes of the *PCSK9* g.NW_011514566_1.406545 T > C SNP were more prevalent than the TT genotype, with a significantly higher milk protein percentage in the CC-genotype individuals than in the TT-genotype individuals (*p* < 0.05). The milk protein percentage in the experimental population with the GG genotype of the *PHLPP1* g.1229 T > G SNP was significantly higher than that in camels with the GT and TT genotypes (*p* < 0.05). The GG and AA genotypes of the *LOC105075649* g.NW_011515666_1-146942 G > A SNP were more prevalent than those carrying the AG genotype, among which there was a significantly higher milk fat percentage in the GG-genotype individuals than in the AG-genotype individuals (*p* < 0.05). Further analysis showed that some SNPs affected multiple milk production traits (Figure 7), indicating pleiotropism, to some degree, or perhaps the biological relatedness of traits.

## 3. Discussion

Phenotypic traits are closely associated with milk production performance. Our previous results are consistent with those of dromedary camels, indicating a strong correlation between breast depth, breast circumference, breast size, and milk production [14,15]. Furthermore, several studies have demonstrated correlations between animal body dimensions and milk yield [16,17]. Genetic and phenotypic correlations between body weight and milk yield were positive in Friesian × Bunaji cows [18]. Previous studies have demonstrated a positive and highly significant correlation among heart girth, body length, body weight, and daily milk yield in buffaloes [19,20]. However, relying on phenotypes and breeding experience to select camels is inefficient and inaccurate.

Milk yield and composition are important variables affecting milk production in dairy animals. Numerous studies have shown that animal milk yield is negatively correlated with milk fat concentration [21,22] and protein concentration [23,24]. The negative correlation between milk yield and nutrient composition makes it very difficult to select dairy camels with both high milk yield and high-quality milk. Genetic selection plays a vital role in improving livestock productivity. Identification and use of molecular markers for camel milk yield and quality will ensure better milk productivity. Mammary gland development and lactation are crucial biological processes regulated by multiple genes and signaling pathways. In our previous studies, transcriptome sequencing revealed that many genes and pathways associated with mammary gland development were differentially expressed between high- and low-milk-yield camels [25,26]. In a recent study, transcriptome sequencing was conducted on the blood of 16 lactating Alashan Bactrian camels from both supplementation and grazing groups. Through weighted gene co-expression network analysis (WGCNA), 1185 genes related to milk production, milk protein, milk fat, and lactose were identified [27]. In the last few years, whole-genome sequencing and resequencing of the camel genome have resulted in the identification of millions of genome-wide SNPs and candidate genes, mainly associated with adaptability to extreme environments, immunity and disease resistance, growth and meat quality, genetic diversity, and the evolutionary origin of camels. However, there has been almost no research on molecular markers for milk production traits.

Rigorous sample selection and high-quality sequencing data were key for subsequent analyses. One major strength of the current study is that all resequencing samples were subjected to a rigorous selection standard, and the sequencing quality was high. The vast majority of SNPs identified are located upstream or downstream of genes or in intergenic regions, and they are mainly T: A > C: G  and C: G > T: A. We used *F_ST_*-*θπ* ratio methods to identify the important candidate genes associated with milk production traits. The intersection of results from the two methods allowed us to identify 264 SNPs in the sequencing population. GO analysis indicated that genes in the selected regions were associated with complex biological processes such as cell signal transduction, ion transport, channel activity, and enzyme activity. Similarly, KEGG pathway enrichment analysis showed that the candidate genes were enriched in multiple signaling pathways, such as oxytocin, estrogen, ErbB, Wnt, mTOR, PI3K-Akt, growth hormone synthesis/secretion/action, and MAPK signaling pathways. The post-natal development of the mammary gland and its function are governed by a hormonal network that mainly comprises steroid hormones (estrogens and progesterone) and peptide hormones of a pituitary origin (prolactin, growth hormone, and oxytocin) [28]. The milk ejection reflex is mediated by oxytocin, which induces myoepithelial cell contraction. Moreover, oxytocin influences milk production by reducing intra-alveolar pressure, reducing the presence of a feedback inhibitor of lactation around the alveoli, and re-establishing normal mammary blood flow [29]. Estrogen influences milk fat synthesis and secretion by regulating mammary gland development and function. In addition, *ErbB2* is a promising candidate gene for regulation of milk protein concentration in dairy cattle [30]. The ErbB3 and ErbB4 signaling pathways play crucial roles in enhancing lactogenesis and differentiation of the mammary glands during pregnancy [31]. Their heterodimers activate PI3K signaling and are candidate regulators of milk protein biosynthesis [32]. The growth hormone is one of the most important hormones, with major effects on growth, reproduction, and milk production [33,34]. The Wnt, mTOR, PI3K-Akt, and MAPK signaling pathways are involved in mammary gland development and lactation [35,36,37,38]. Therefore, genes in the above-mentioned signaling pathways are important candidates for validation.

To investigate the molecular markers affecting milk production traits, we performed genotyping and association analyses of the top 123 SNPs associated with important pathways and strongly selected regions identified in the expanded experimental population. As shown in Figure 7, 13 SNPs were identified as being significantly associated with milk yield, whereas 18 SNPs were associated with milk composition. Most of these polymorphic sites were located in the coding regions of the genome. However, five and two important mutation sites were found in the non-coding regions (promoter and introns) of *CSN2* and *CSN3*, respectively. Polymorphisms in non-coding regions may underlie the differences in expression between certain alleles [10,39]. Several of these SNPs exhibited significant pleiotropic effects on multiple milk production traits. Biological functions of candidate genes in regions harboring selection signatures have been identified in previous studies. Increased cow milk fat production may be positively regulated by *NR4A1*, which is activated by the transcription factor SP4 [40]. *ADCY8* is expressed in brain regions that control energy homeostasis and nutrition [41]. A study on reported quantitative trait loci in the *ADCY8* gene region of animal genomes showed an association with milk yield and milk fat percentage [42,43]. ROR2 is a receptor for Wnt5a signaling that regulates branching, differentiation, and actin cytoskeletal dynamics within the mammary epithelium [44]. NRG3, a neuronal enriched growth factor, promotes early mammary morphogenesis [45]. *IGF1R* is a multifunctional gene that is not only an important candidate gene for milk traits in cattle but also closely related to pig production performance and growth traits in cattle and sheep [46,47,48]. *PCSK9* plays a critical role in the regulation of cholesterol homeostasis [49]. The main functions of *CRKL* are related to cell signal transduction and proliferation [50]. Tmem44 is a putative transmembrane protein [51]. Polymorphisms in *DMD* are associated with growth and body size traits in pigs [52]. PPARG is a significant regulator of lipid homeostasis, and its coding gene is closely related to goat milk production traits [53]. *PHLPP1* is related to production and growth traits [54].

*CSN2* and *CSN3* are associated with milk production traits in *Camelus dromedarius* and Kazakh Bactrian camels [8,55]. Similarly, verification through SNP analysis showed that seven SNP loci were distributed in *CSN2* and *CSN3*, all of which were related to milk fat and protein percentages in Junggar Bactrian camels. The correlation between *CSN2* and *CSN3* genotypes and the percentages of milk fat and protein may be due to varying the mRNA stability and transcription rates, which in turn influences protein expression and the resultant phenotype [7,8].

Given that the domestic Bactrian camel reference genome has not yet been assembled at the chromosome level, we cannot currently delineate whether the above-mentioned related SNPs are in linkage disequilibrium on the chromosome. These findings suggest important candidate genes for breeding high-quality Bactrian camels. Therefore, we speculated that the transcription and translation of these candidate genes may affect milk production traits by altering mammary cell proliferation, milk fat synthesis, and milk protein synthesis. Nevertheless, further studies are needed to verify the molecular functions of these candidate genes at the cellular and protein levels.

## 4. Methods

### 4.1. Animals and Management

The experimental camels were obtained from local herdsman who traditionally graze 98 Junggar Bactrian lactating she-camels on native pasture without feed supplements. These female camels are the breeding offspring of Junggar Bactrian camels and have been raised and bred by local herdsmen for many years. The whole-genome resequencing (WGRS) population consisted of 24 camels with unknown relationships among them, including 12 high- and 12 low-milk-productivity lactating she-camels from Fuhai County, Altay Prefecture, Xinjiang Uygur Autonomous Region, northwest China (Figure 8A). Camels were selected for whole-genome sequencing in October 2019. These camels were located at two sampling sites (Figure 8B, areas 1 and 2), with six high-yield and six low-yield Bactrian camels being selected (Figure 8C). Camels at each sampling site were exposed to the same feeding patterns and management methods. All the above-mentioned camels were 9–10-years-old at parity three or four (Appendix A). An SNP genotyping experimental population comprised of healthy lactating she-camels (*n* = 521, parity 1–2, 4–5 years old) from a Camel Dairy Farm located in Jimunai County was also included in the validation study (Figure 8B, area 3). All lactating she-camels were selected from randomly chosen healthy camels.

### 4.2. Milk Yield and Composition Recording

Measurement of milk production began after the 45th day post-partum to allow the young camels to consume milk. On the night preceding the milk yield recording, young camels were separated from their dams at 22.00 h. The next morning, at 06.00 h, young camels were allowed to suckle their dams for a period that did not exceed 3 min. The amount of milk consumed by young camels was not included in the calculation. Lactating she-camels were milked three times a day during the peak of the lactation period only, whereas they were milked twice a day through the rest of the lactation period. Milk yield was weighed after each milking. Therefore, milk production was recorded daily. The actual total milk yield was divided by 300 to calculate the mean daily milk yield. Finally, two groups were identified: a high milk-yielding group (>2.75 kg/d) and low milk-yielding group (<1.96 kg/d) based on daily milk production. The data from the entire lactation period (300 d) of the parity of each she-camel were used as the individual milk yield phenotypes. Camel milk was collected at the beginning, middle, and end of each month. The morning and evening milk samples were mixed immediately after collection and transported to the laboratory in an ice bath. Milk samples from each she-camel were not pooled together and were instead detected individually. Samples were analyzed for solids-not-fat, fat, protein, and lactose using a fully automated milk composition analyzer (UL40BC, Youchuang, Hangzhou, China). Each measurement was repeated three times, and the average value was calculated. Results are shown as the mean ± standard deviation (SD), and the statistical significance was highlighted by a Student’s *t*-test using SPSS (version 26.0). Plots were constructed using R software (version 4.2.0).

### 4.3. Sample Collection

Blood samples were collected from camels through jugular venipuncture into EDTA-coated vacutainer tubes without apparent discomfort. Blood samples were cryopreserved until DNA extraction and analysis. Milk samples were analyzed for fat, protein, lactose, and solid non-fat (SNF) content using a milk composition analyzer apparatus (UL40BC, Hangzhou, China). The micromorphological study of the mammary gland was carried out comparatively on high- and low-yielding Bactrian she-camels.

### 4.4. Sample Processing and Microscopic Observations

Mammary gland micromorphologies in high- and low-yielding Bactrian camels were analyzed comparatively. Tissue slices of the mammary glands were stained with hematoxylin and eosin (HE) following protocols from a previous study [56]. Histological images were captured under a microscope (Eclipse E100, Nikon, Japan).

### 4.5. DNA Library Construction, Sequencing, Read Mapping, and Quality Control

Genomic DNA was extracted from camel blood samples using a Puregene Tissue Core Kit A (Qiagen) following the manufacturer’s protocol. The quality and integrity of the DNA were assessed using the A260/280 ratio and 1.2% agarose gel electrophoresis, respectively. To prepare the sequencing library, high-quality genomic DNA from each individual was used to construct a paired-end sequencing library with an insert size of 400 bp, which was then sequenced on an Illumina NovaSeq platform (Illumina, San Diego, CA, USA). FastQC version 0.11.7 was used to visualize the quality of the sequence reads. The predicted average sequencing depth per sample was 15×. The sequencing depth for the samples included in this analysis are provided in Appendix A. To ensure the quality of sequencing data, raw low-quality paired reads were removed through a series of steps: (1) reads with 3′-end adapter contamination were removed by Adapter Remova (version 2) [57]; (2) sequences with paired-end read lengths ≤ 50 bp were filtered out; and (3) the sliding window method was used to filter sequencing data based on quality, with the window size set to 5 bp and the step size set to 1 bp. Each time a base moved forward, the average Q value of the window was calculated from 5 bases. If the Q value of the last base was ≤2, then only the bases before this position were retained, and if the average Q value of the window was ≤20, only the second-to-last base in the window and the bases before it were retained.

Filtered reads were then mapped to the *C. bactrianus* reference genome (Ca_bactrianus_MBC_1.0), available from National Center for Biotechnology Information (NCBI) database, using Burrows–Wheeler Aligner software (Version 0.7.12) [58], and the parameters ‘mem -t 4 -k 32 -M’ SNP detection was performed using the UnifiedGenotyper program in the GenomeAnalysis TK (version 3.8) software package [59]. To ensure the accuracy of an SNP site, high-quality variants were selected among raw SNPs based on the following quality scores: Fisher test of strand bias (FS) < 60, mapping quality (MQ) > 40, quality depth (QD) > 2, haplotype score < 13, MQRankSum > −12.5, approximate read depth (DP) > 5, and ReadPosRankSum > −8.0.

### 4.6. Identification of Selective Sweep Detection of Candidate Gene

A genome-wide selective sweep detection analysis was performed on a genome-wide scale, and a detection strategy combining the population genetic differentiation index (*F_ST_*) and nucleotide diversity (*θπ*) was used to apply jointly identified regions as candidate selected regions for extraction of variant site information in the corresponding regions. The *F_ST_* values were as follows:(1)$$F_{ST=}\frac{MSP-MSG}{MSP+(n_{c}-1)\;MSG}$$
where *MSP* is the mean square error within the population, *MSG* represents the mean square error between two populations, and n_c_ represents the average sample size of the entire population after correction [60,61]. The *θπ* ratio was *θ_H_*/*θ_L_* log_10_ transformed. We used high-quality SNPs, as described above, for further analysis to identify selection signals that led to changes in milk production in camels. The *F_ST_* value and *θπ* ratio were calculated in sliding 20-kb windows with a step size of 5 kb between camels. Selection mapping methods were extracted from the 1% extreme left and right tails of the distribution by applying cutoffs of *θπ* [62]. The regions detected using both methods were identified as the final selective mapping methods, considering the intersection of the windows of the top 1% of the two methods as candidate regions.

### 4.7. Signatures of Selection Annotation and Gene Functional Enrichment Analysis

To further explore the functions of the genes in the selected regions, we performed GO enrichment (http://www.geneontology.org/, accessed on 11 September 2023) and KEGG pathway enrichment (http://www.genome.jp/kegg/pathway.html, accessed on 11 September 2023) analyses. Candidate gene functions were primarily determined using information from the NCBI database (http://www.ncbi.nlm.nih.gov, accessed on 13 September 2023).

### 4.8. SNP Genotyping and Association Analysis

We selected genes located in important pathways and relevant regions to genotype the top 123 SNPs. Genotyping of 123 SNPs was performed using the Sequenom MassARRAY platform based on a previously published method (Appendix A) [63,64]. PCR reactions and single-base extension primers were designed using the MassARRAY Assay Design SUITE (Version 2.0) (Appendix A). The full results from all SNP primers are shown in Appendix A. The PCR conditions are shown in Appendix A. Purified primer extension reaction products were spotted onto a 384-well spectrophotometer using a MassARRAY Nanodispenser (Sequenom), and the genotypes were determined using MALDI-TOF-MS. A PCR-SSCP analysis was performed to analyze mutations in important candidate genes. Correlations between milk production traits and different genotypes were determined using linear regression analysis with a general linear model in the validation she-camel population. The generalized linear model (GLM) used for camel milk production traits was as follows:Y = μ + P + A + G + e(2)
where Y are milk production traits, μ is the overall population mean, P is a fixed effect of parity, A is a fixed effect on the age of female camels, G is the fixed effect of a genotype or haplotype, and e is the random residual error. The phenotype values are presented as the mean ± SD, and all statistical tests were considered statistically significant at *p* < 0.05. The R codes used for the analyses are presented in Appendix A. The genotype frequency was calculated by dividing the number of individuals with a given genotype by the total number of samples.

## 5. Conclusions

The results of this study suggested that the chemical content of camel milk was lower in high-yielding camels than in low-yielding Bactrian camels. We verified 123 SNPs in the expanded camel population and observed 13 SNPs with significant genetic associations with milk yield and 18 SNPs with significant genetic associations with milk composition percentages. From the candidate genes identified, eight (*NR4A1*, *ADCY8*, *ROR2*, *NRG3*, *IGF1R*, *PCSK9*, *CRKL*, and *LOC105075649*) were linked with milk yield and 12 (*ADCY8*, *TMEM44*, *DMD*, *PPARG*, *ROR2*, *CSN2*, *CSN3*, *CDH7*, *PCSK9*, *PHLPP1*, *LOC105075649*, and *CDH7*) were related to milk composition percentages in lactating Bactrian camels. After further validation, these results will be useful for marker-assisted breeding.

## Figures and Tables

**Figure 1 ijms-25-07836-f001:**
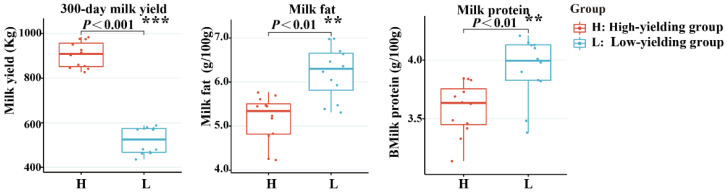
Differential physiological and biochemical indexes in high- and low-yielding camels. ** *p* < 0.01; *** *p* < 0.001; NS, not significant.

**Figure 2 ijms-25-07836-f002:**
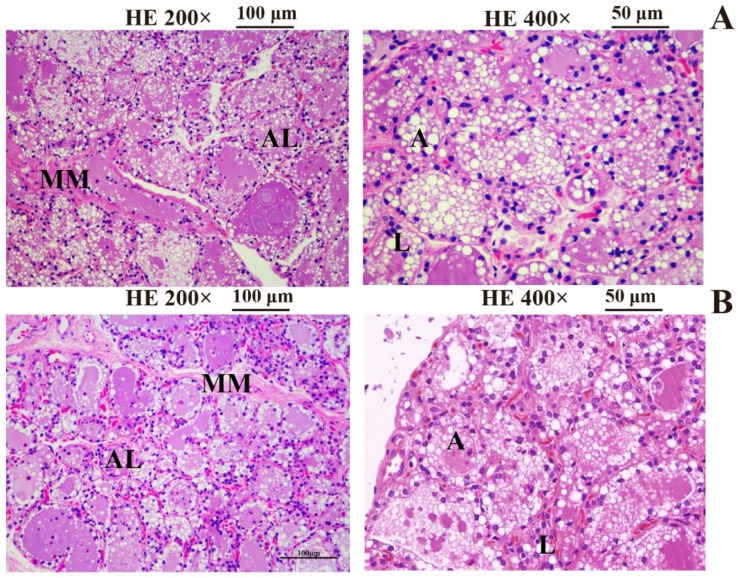
Comparison of microscopic morphology of mammary glands in high- and low-yield camels. (**A**) Mammary glands of low-yield camels; (**B**) mammary glands of high-yield camels. AL, acinar lumen; MM, mammary mesenchyme; A, acinar epithelial cells; L, lipid droplets.

**Figure 3 ijms-25-07836-f003:**
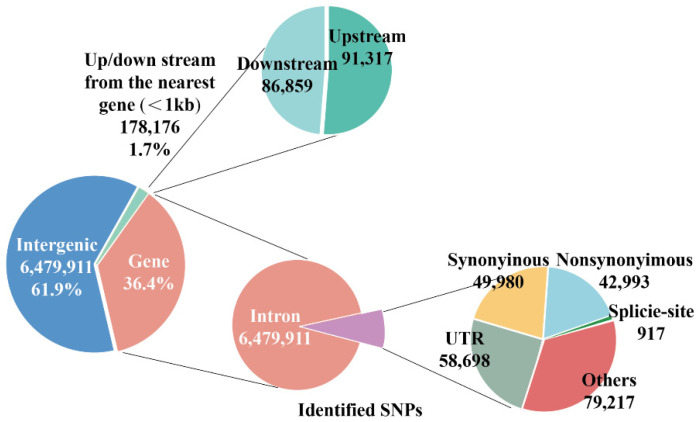
Classification of identified single-nucleotide polymorphisms.

**Figure 4 ijms-25-07836-f004:**
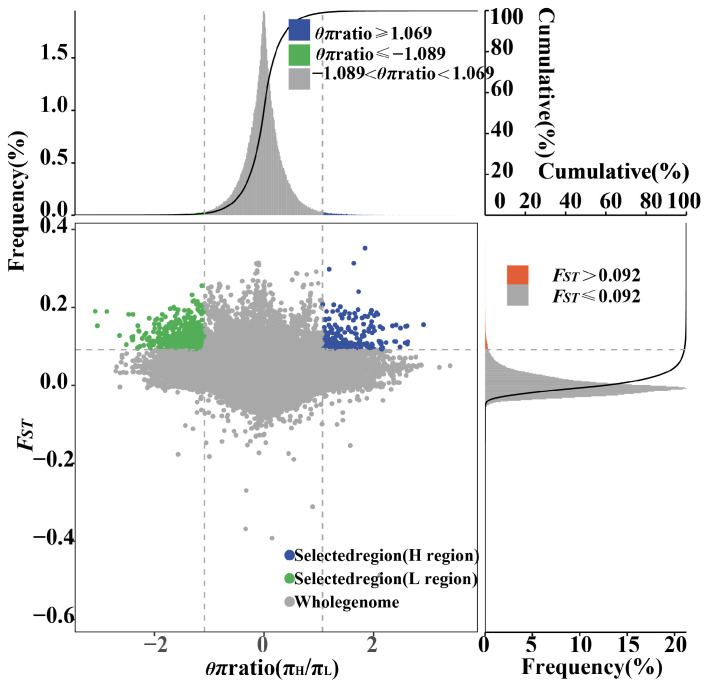
Genomic regions with strong selective signals in high- and low-yielding camels. *F_ST_* and *θπ* ratios were calculated using a sliding window analysis with a window size of 20 kb and a step size of 5 kb. Data points in blue corresponding to the top 1% of *F_ST_* and the top 1% of the *θπ* ratio distribution are genomic regions under selection in high-yield camels. Data points in green corresponding to the top 1% of the *F_ST_* and the top 1% of the *θπ* ratio distribution are genomic regions under selection in low-yield camels.

**Figure 5 ijms-25-07836-f005:**
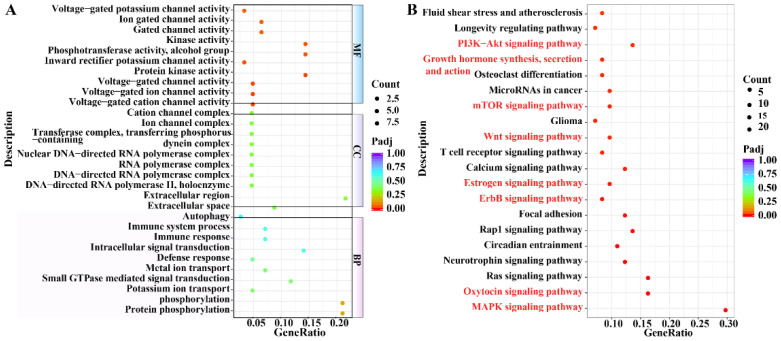
Functional enrichment and pathway analyses. (**A**) Top 10 terms in each GO category ranked according to their statistical significance; (**B**) top 20 enriched KEGG pathways.

**Figure 6 ijms-25-07836-f006:**
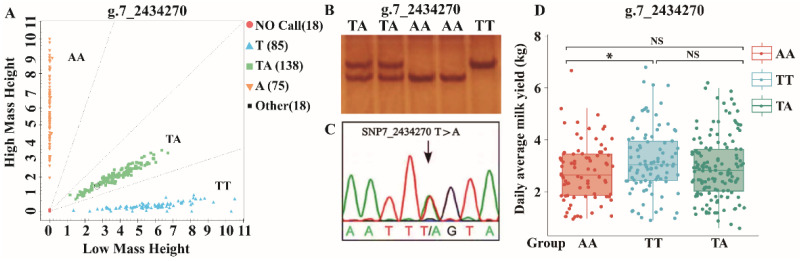
Genotyping of Bactrian camel *ROR2* g.7_2434270 T > A. (**A**) *ROR2* g.7_2434270 T > A; (**B**) PCR-SSCP detection results of *ROR2* g.7_2434270 T > A; (**C**) single peak in the sequencing peak map also indicated successful mutation creations on the *ROR2* g.7_2434270 T > A; and (**D**) effect of *ROR2* g.7_2434270 genotype on milk production. * *p* < 0.05; NS, not significant.

**Figure 7 ijms-25-07836-f007:**
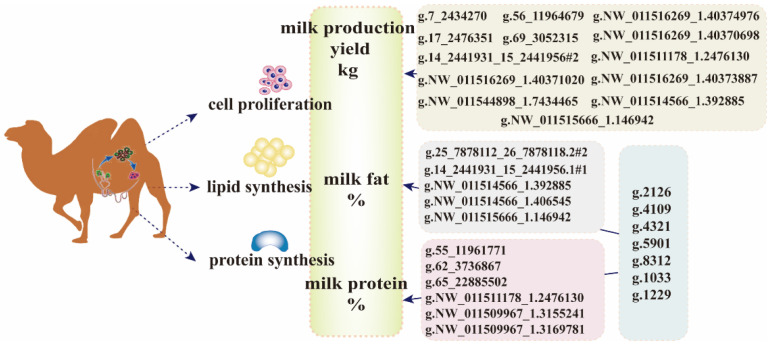
The relationship between SNPs and milk production and milk composition traits in Bactrian camels.

**Figure 8 ijms-25-07836-f008:**
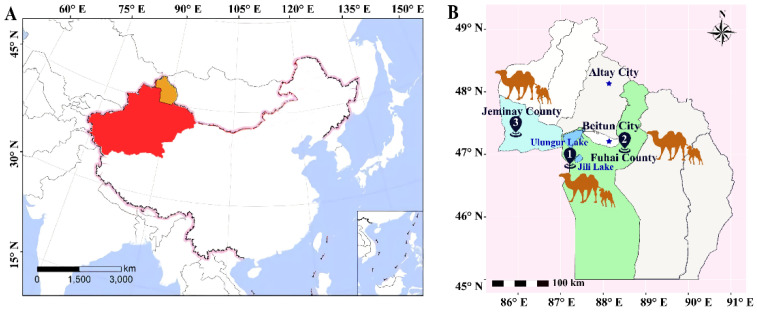
Geographical locations of the samples. (**A**) North-western China; (**B**) Xinjiang Altay region; (**C**) sampling camel photos.

**Table 1 ijms-25-07836-t001:** Comparison of milk traits between high- and low-yielding camels.

Milk Traits	High-Yielding Camels			Low-Yielding Camels			*p*-Value	Sign.
	H1–H12			L1–L12				
	Mean ± SD	Maximum	Minimum	Mean ± SD	Maximum	Minimum		
300-day milk yield (kg)	904.83 ± 57.80	983.68	827.16	520.14 ± 59.05	587.35	435.31	*p* < 0.001	***
Fats (g/100 g)	5.16 ± 0.51	5.77	4.23	6.21 ± 0.59	6.99	5.31	*p* < 0.01	**
Solids non-fat (g/100 g)	8.91 ± 0.37	9.34	8.05	9.10 ± 0.45	10.0	8.51	0.287	NS
Protein (g/100 g)	3.59 ± 0.22	3.84	3.14	3.93 ± 0.27	4.21	3.38	*p* < 0.01	**
Ash specification (g/100 g)	0.72 ± 0.03	0.77	0.66	0.74 ± 0.03	0.80	0.70	0.186	NS
Lactase (g/100 g)	4.65 ± 0.20	5.00	4.31	4.74 ± 0.17	5.15	4.55	0.228	NS

Sign., level of significance; ** *p* < 0.01; *** *p* < 0.001; NS, not significant.

**Table 2 ijms-25-07836-t002:** Association results between milk production yield and different genotypes of the SNPs.

SNP_ID	Region	Gene	Full Name	Mutation Type	Genotype Frequency	Daily Average Milk Yield (kg)
17_2476351	exon	*NR4A1*	Nuclear receptor 4 group A1	GG *	0.52	3.12 ± 1.23 ^a^
				TT	0.48	2.81 ± 1.17 ^b^
56_11964679	exon	*ADCY8*	Adenylate Cyclase 8	CC	0.82	2.93 ± 1.17 ^b^
				CT	0.16	2.97 ± 1.27 ^b^
				TT *	0.02	4.63 ± 1.28 ^a^
7_2434270	exon	*ROR2*	Receptor tyrosine kinase-like orphan receptor 2	AA	0.26	2.77 ± 1.13 ^a^
				TA	0.44	2.92 ± 1.21 ^ab^
				TT *	0.30	3.20 ± 1.25 ^b^
14_2441931_15_2441956#2	exon	*NRG3*	Neuregulin 3	CC *	0.76	3.03 ± 1.23 ^a^
				TC	0.08	2.36 ± 0.89 ^b^
				TT	0.16	2.95 ± 1.19 ^ab^
69_3052315	exon	*IGF1R*	Insulin-like Growth Factor-I Receptor	CC	0.76	2.89 ± 1.21 ^ab^
				TC *	0.23	3.22 ± 1.23 ^a^
				TT	0.01	2.81 ± 0.89 ^b^
NW_011516269_1.40371020	exon	Unknown	Unknown	CC *	0.37	3.09 ± 1.17 ^a^
				TC	0.12	2.63 ± 1.05 ^ab^
				TT	0.51	2.60 ± 1.10 ^b^
NW_011516269_1.40373887	exon	Unknown	Unknown	GG	0.51	2.60 ± 1.10 ^a^
				GT	0.36	2.88 ± 1.21 ^ab^
				TT *	0.13	3.24 ± 0.95 ^b^
NW_011544898_1.7434465	exon	Unknown	Unknown	CC	0.06	2.07 ± 0.80 ^a^
				TC	0.35	2.73 ± 1.20 ^ab^
				TT *	0.58	2.90 ± 1.11 ^b^
NW_011514566_1.392885	exon	*PCSK9*	Preprotein transferase enzyme subtilin lysin 9	AA *	0.36	3.05 ± 3.05 ^a^
				AG	0.48	2.70 ± 2.70 ^ab^
				GG	0.16	2.46 ± 2.46 ^b^
NW_011511178_1.2476130	exon	*CRKL*	CRK like proto-oncogene, adaptor protein	GG	0.23	2.47 ± 0.91 ^a^
				GT *	0.49	2.96 ± 1.21 ^b^
				TT	0.28	2.72 ± 1.07 ^ab^
*NW_011515666_1.146942*	exon	*LOC105075649*	*LOC105075649*	AA	0.16	2.73 ± 1.19 ^ab^
				AG *	0.43	3.05 ± 1.13 ^a^
				GG	0.41	2.54 ± 1.10 ^b^
NW_011516269_1.40374976	exon	Unknown	Unknown	CC	0.51	2.60 ± 1.12 ^a^
				CG	0.36	2.89 ± 1.19 ^ab^
				GG *	0.13	3.27 ± 0.97 ^b^
NW_011516269_1.40370698	exon	Unknown	Unknown	AA *	0.13	3.24 ± 0.95 ^a^
				GA	0.36	2.87 ± 1.22 ^ab^
				GG	0.51	2.63 ± 1.15 ^b^

Note: phenotype values are presented as the mean ± the standard error. In the same column, significant differences are labeled with different lowercase letters (*p* < 0.05). The asterisk (*) denotes the predominant genotype.

**Table 3 ijms-25-07836-t003:** Association results between milk composition percentages and different genotypes of the SNPs.

SNP ID	Region	Gene	Full Name	Mutation Type	Genotype Frequency	Milk Fat (g/100 g)	Milk Protein (g/100 g)
55_11961771	exon	*ADCY8*	Adenylate cyclase 8	AA	0.47	5.48 ± 0.29	3.74 ± 0.14 ^a^
				GA	0.43	5.40 ± 0.25	3.73 ± 0.12 ^a^
				GG *	0.10	5.49 ± 0.28	3.94 ± 0.19 ^b^
25_7878112_26_7878118.2#2	exon	*TMEM44*	Transmembrane protein 44	CC *	0.86	5.50 ± 0.28 ^a^	3.83 ± 0.11
				TC	0.02	5.40 ± 0.32 ^ab^	3.75 ± 0.20
				TT	0.12	5.37 ± 0.25 ^b^	3.69 ± 0.14
62_3736867	exon	*DMD*	Dystrophin	AA	0.61	5.49 ± 0.29	3.74 ± 0.19 ^a^
				AG	0.34	5.45 ± 0.28	3.73 ± 0.14 ^a^
				GG *	0.05	5.57 ± 0.28	3.90 ± 0.22 ^b^
65_22885502	exon	*PPARG*	Peroxisome proliferator activated receptor gamma	GA	0.03	5.40 ± 0.27	3.73 ± 0.13 ^a^
				GG *	0.97	5.48 ± 0.27	3.90 ± 0.58 ^b^
14_2441931_15_2441956.1#1	exon	*ROR2*	Receptor tyrosine kinase like orphan receptor 2	GC	0.09	5.36 ± 0.29 ^a^	3.69 ± 0.12 ^a^
				GG *	0.91	5.50 ± 0.25 ^b^	3.74 ± 0.20 ^b^
CSN2_2126	Promoter	*CSN2*	β-casein	GG	0.44	5.44 ± 0.42 ^b^	3.61 ± 0.24 ^b^
				GA	0.39	5.31 ± 0.53 ^b^	3.59 ± 0.29 ^b^
				AA *	0.17	5.86 ± 0.37 ^a^	3.94 ± 0.31 ^a^
CSN2_4109	Intron	*CSN2*	β-casein	GG *	0.45	5.79 ± 0.28 ^a^	4.05 ± 0.27 ^a^
				GT	0.37	5.25 ± 0.35 ^b^	3.57 ± 0.24 ^b^
				TT	0.28	5.34 ± 0.41 ^ab^	3.63 ± 0.32 ^ab^
CSN2_4321	Intron	*CSN2*	β-casein	GG *	0.09	5.84 ± 0.21 ^a^	3.97 ± 0.27 ^a^
				GT	0.38	5.29 ± 0.59 ^b^	3.58 ± 0.28 ^b^
				TT	0.53	5.28 ± 0.57 ^b^	3.47 ± 0.30 ^b^
CSN2_5901	Intron	*CSN2*	β-casein	AA *	0.16	5.77 ± 0.35 ^a^	3.88 ± 0.26 ^a^
				AG	0.38	5.24 ± 0.42 ^b^	3.56 ± 0.33 ^b^
				GG	0.46	5.26 ± 0.49 ^b^	3.48 ± 0.29 ^b^
CSN2_8312	Intron	*CSN2*	β-casein	GG	0.13	5.19 ± 0.52 ^b^	3.41 ± 0.37 ^b^
				GA	0.42	5.25 ± 0.47 ^b^	3.52 ± 0.36 ^b^
				AA *	0.44	5.76 ± 0.32 ^a^	3.93 ± 0.34 ^a^
CSN3_1033	Intron	*CSN3*	κ-casein	CC	0.42	5.34 ± 0.24 ^b^	3.49 ± 0.26 ^b^
				CA	0.43	5.22 ± 0.30 ^b^	3.42 ± 0.31 ^b^
				AA *	0.15	5.82 ± 0.32 ^a^	3.89 ± 0.35 ^a^
CSN3_1229	Intron	*CSN3*	κ-casein	GG *	0.23	5.83 ± 0.43 ^a^	3.96 ± 0.37 ^a^
				GT	0.26	5.26 ± 0.27 ^b^	3.40 ± 0.28 ^b^
				TT	0.51	5.32 ± 0.38 ^b^	3.42 ± 0.36 ^b^
NW_011509967_1.3155241	exon	*CDH7*	Cadherin 7	AA	0.33	5.38 ± 0.24	3.65 ± 0.18 ^a^
				GA *	0.43	5.49 ± 0.29	3.73 ± 0.19 ^b^
				GG	0.27	5.41 ± 0.25	3.64 ± 0.17 ^a^
NW_011514566_1.392885	exon	*PCSK9*	Proprotein convertase subtilisin/kexin type 9	AA	0.36	5.38 ± 0.26 ^a^	3.66 ± 0.19
				AG	0.48	5.45 ± 0.26 ^ab^	3.68 ± 0.18
				GG *	0.16	5.55 ± 0.28 ^b^	3.72 ± 0.18
NW_011515224.1. 967735	exon	*PHLPP1*	PH domain and leucine rich repeat protein phosphatase 1	GG	0.23	5.43 ± 0.24	3.69 ± 0.19 ^a^
				GT	0.51	5.41 ± 0.25	3.64 ± 0.16 ^b^
				TT *	0.26	5.52 ± 0.27	3.74 ± 0.21 ^a^
NW_011515666_1.146942	exon	*LOC105075649*	LOC105075649	AA	0.16	5.43 ± 0.27 ^ab^	3.67 ± 0.19
				AG	0.41	5.37 ± 0.22 ^a^	3.63 ± 0.15
				GG *	0.43	5.52 ± 0.30 ^b^	3.71 ± 0.19
NW_011514566_1.406545	exon	*PCSK9*	Proprotein convertase subtilisin/kexin type 9	CC *	0.15	5.53 ± 0.28 ^a^	3.71 ± 0.18
				CT	0.47	5.46 ± 0.26 ^ab^	3.68 ± 0.18
				TT	0.38	5.39 ± 0.26 ^b^	3.66 ± 0.19
NW_011509967_1.3169781	exon	*CDH7*	Cadherin 7	AA	0.35	5.44 ± 0.26	3.66 ± 0.18 ^ab^
				GA *	0.44	5.46 ± 0.28	3.72 ± 0.19 ^a^
				GG	0.21	5.42 ± 0.25	3.64 ± 0.17 ^b^

Note: phenotype values are presented as the mean ± the standard error. In the same column, significant differences are labeled with different lowercase letters (*p* < 0.05). The asterisk (*) denotes the predominant genotype.

## Data Availability

The datasets analyzed during the current study are available from the corresponding author on reasonable request. All sequencing data are available through the NCBI Sequence Read Archive under the accession number PRJNA876601.

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
