# Peer review of "Whole-Genome Resequencing Analysis of the Camelus bactrianus (Bactrian Camel) Genome Identifies Mutations and Genes Affecting Milk Production Traits"

_ijms, 2024, doi:10.3390/ijms25147836_

Round 1

Reviewer 1 Report

Comments and Suggestions for Authors

The manuscript entitled “Whole-genome resequencing analysis of the Camelus bactrianus (Bactrian camel) genome identifies mutations and genes affecting milk production traits” is a clearly written article, which mainly aims to identify potential molecular markers impacting milk production in camels.

The manuscript authors have made a considerable effort to perform this study. However, the statistical analysis of data requires critical modifications. Please check the details below.

Introduction section could be reduced and be focused mainly on camels.

All software and programs used should be mentioned in detail.

Conclusion is well-written and particularly highlighting what is the most utilitarian side of the findings. Thanks to authors.

Comments on the Quality of English Language

It is written in good English language. Few errors have to be corrected

Author Response

Response to Reviewer Comments

Dear reviewer:

Thanks for your questions and suggestions. Following your professional comments, we have re-edited our manuscript throughout carefully and addressed all the comments in the notes below.

Please note that our responses to the reviewer’s comments are shown in red font, and line numbers mentioned here are shown in the manuscript.

Point 1: Line 118: nutritional composition of camel milk negatively correlated with milk productionto be chemical composition of camel milk is dependent on variations in milk yield.

Response 1: Thanks for your comments. We apologize for the unclear description of this sentence. We have rephrased this sentence as the nutritional content of high-yielding camel milk is lower than that of low yielding camels. (Lines 117-118)

Point 2: Line 126: of camels to be of she-camels.

Response 2: We have changed camels to camels. (Line 125)

Point 3: Line 127:"breast ducts"to be udder ducts.

Response 3: We have changed breast ducts to udder ducts. (Line 126)

Point 4: Line 130:"t high-yielding"to be the high-yielding

Response 4: Thanks for reviewer’s reminder. We have made modifications. (Line 129)

Point 5: Line 428:"98 Junggar Bactrian camels".....Are they she-males or both males and females?

Response 5: Thanks for reviewer’s comments. 98 Junggar Bactrian camels are all female camels. At this stage, all male camels freely feed in the wild mountains. “98 Junggar Bactrian lactating she-camels on native pasture without feed supplements.”

Point 6: Line 433:"These camels were located in two sampling sites"....I got confused about sampling. What do you mean by located in two sampling sites? Is this a single flock divided into 2 areas? Or they are 2 different flocks?

Response 6: Thanks for reviewer’s comments. Firsty, we have added Figure 8 to the manuscript (Lines 438-441). We chose sequencing samples based on factors such as age and milk production. In a pastoral area, the number of suitable sample camels is small. We chose two pastoral areas to increase the number of camels and improve the accuracy of the experiment.

This is a single flock divided into 2 areas due to limitations in grassland conditions. These female camels are the breeding offspring of the Junggar Bactrian camel and have been raised and bred by local herdsmen for many years (Lines 426-428).

Point 7: Line 437:"lactating camels"to be lactating she-camels.

Response 7: As suggested by the reviewer, we have modified it as required. (Line 430)

lactating camels→lactating she-camels.

Point 8: Line 438:"Dairy Farm in Jimunai County"to be Dairy Farm located in Jimunai County.

Response 8: As suggested by the reviewer, we have modified it as required. (Line 437)

Dairy Farm in Jimunai County→Dairy Farm located in Jimunai County.

Point 9: Line 438:"included in the study"to be included in the validation study.

Response 9: As suggested by the reviewer, we have modified it as required. (Line 438)

included in the study→included in the validation study.

Point 10: Line 443:"from mother camels"to be from their dams

Response 10: As suggested by the reviewer, we have modified it as required. (Line 447)

from mother camels→from their dams

Point 11: Line 445:"During the peak lactation period, mother camels milk three times a day, whereas during other periods, they milk twice a day"to be Lactating she-camels were milked three times a day during the peak of lactation period only, whereas they were milked twice a day along the entire lactation period.

Response 11: As suggested by the reviewer, we have modified it as required. (Lines 449-451)

Point 12: Line 446:At which week she-camels reached the lactation peak?

Response 12: The peak lactation period of the Junggar Bactrian camel is from early July to the end of September each year. This is our study on milk production data during lactation weeks.

Point 13: Line 450:"each camels"to be each she-camel.

Response 13: As suggested by the reviewer, we have modified it as required. (Line 454)

Point 14: Line 462: "The mammary gland tissue was collected" to be the he micromorphological study of the mammary gland was carried out comparatively on high-and low-yielding Bactrian she-camels.

Response 14: Thanks for your suggestions. We have added this paragraph to the manuscript. (Lines 466-467)

Point 15: Line 478:which software was used for assessing a per-base sequence quality?

Response 15: Thanks for your comments. We performed a per base sequence quality check using the software FastQC (v0.11.7). (Line 480)

Point 16: Line 482:What was the obtained size(depth)of raw sequences per sample?

Response 16: Thanks for your comments. We have added this sentence “The sequencing depth for the samples included in this analysis are provided in Table S3.” Lines (481-482)

Point 17: Line 485:What were the criteria applied to minimize nonindependences among adjacent windows and ensure sufficient resolutions?

Response 17: Thanks for your comments. Our analysis chooses to clear a window of 20kb, with a step of 5kb (5 kb forward each time), i.e., there is an overlap between the windows.

Overlap Size: One way to reduce edge effects and ensure continuity between adjacent windows is by introducing an overlap between them. The optimal overlap size depends on the specific requirements of the analysis and the characteristics of the data but is chosen to balance reducing nonindependences with computational efficiency.

Point 18: Line 502-503: It is great to calculate both genetic differentiation(FST) and nucleotide diversity (θπ). Why authors did not compute the absolute genetic divergence(dxy)between two categories of milk production performance?

Response 18: Thanks for your comments. We focus more on the genetic structure and diversity between or within populations, rather than directly comparing the absolute genetic distance between two groups of individuals with different production performance.

Our analysis, which compares genomic regions fixed by selection on the genomes between the two groups, is looking at the specific distribution of differences, and the current use of FST, pi, would be more appropriate.

FST and θ π can reflect these aspects of information well, while dxy may not be the main focus of this study. FST, θ π, and dxy each have their unique uses and values in genetics.

dxy is more concerned with measuring the extent of specific genetic variation between two sequences or populations, rather than the distribution of variation or the extent of fixed differences.

Although dxy can provide absolute values of genetic differences between two groups of individuals, it may not fully reflect the genetic structure and diversity information revealed by FST and θ π. We believe that calculating FST and θπ is sufficient to reflect the key issues in the study, without the need for additional calculations of dxy.

DOl: 10.1038/ng.2811

Point 19: Line 544-545:"This study suggests that milk yield is negatively correlated with milk fat and milk protein concentrations in Bactrian camels".... Personally, I do not think that this is a novel discovery of the study. Negative genetic correlations between milk yield and composition percentages are facts in livestock.

Response 19: Thanks for your comments. We have re-edit the content of this sentence separately according to the suggestion. (Lines 552-553)

Point 20: Line 547:"milk composition to be milk composition percentages

Response 20: Thanks for your suggestions. we have modified it as required. (Lines 263, 454)

milk composition→milk composition percentages

Point 21: "determined"to be highlighted

Response 21: Thanks for your comments. we have modified it as required. (Line 45)

determined→highlighted

Point 22: Line 548-549:it could be better to separate the genes affecting milk yield from those associated with milk composition.

Response 22: Thanks for your comments. We have re-edit the content of this sentence separately according to the suggestion. (Lines 556-560)

Point 23: Line 535: The mathematical model applied for studying milk production traits is not correct. Some important fixed effects were not included in the model, which probably make the obtained results not accurate. I'd like to ask the authors to repeat the statistical analysis once again, involving the fixed effects of parity (two categories:3d and 4th), age of she-camels (two categories:9 and 10 years old) and geographical area or sampling sites (two categories: area 1 and area 2). What about different milk production performance, (high-and low-yield groups)?

Response 23:       Thanks for your professional comments. During the editing process of the manuscript, age and parity effects were not included as two parameters. However, our statistical analysis process and data are correct. We have checked and modified the mathematical model by adding fixed effect parameters for age and parity. Camels from sampling sites 1 and 2 were used for whole genome resequencing analysis. All samples for association analysis were collected at sampling site 3, so there is no need to add fixed effects at sampling site.

Reviewer 2 Report

Comments and Suggestions for Authors

In my opinion, the manuscript under the title Whole-genome resequencing analysis of the Camelus bactrianus (Bactrian camel) genome identifies mutations and genes affecting milk production traits brings a lot of new information regarding camel dairy use and should be considered by the journal for publication. The authors took great care in writing an introduction in which they included a sufficient description and stated the purpose of the paper. This is followed by the results, which were developed correctly, as was the discussion, but I have a few comments in the methodology and the design of the experiment itself. In my opinion, it would be worthwhile to consider studying one more additional group of camels with average milk yield for comparison of values, so as to make sure that the results obtained in the study were not random.

The methodology lacks very important information regarding sampling - when how many times with what intervals, whether always at the same time, in the morning/evening if only in the morning then whether this did not affect the results obtained, how it related to the stage of lactation, whether the milk was preserved, how milk yield was determined and by what methods the basic composition of the milk was determined.

In the literature list, the second item has an incorrect journal name.

Author Response

Response to Reviewer Comments

Dear reviewer:

Thanks for your professional advices. Following your suggestions, we have re-edited our manuscript throughout carefully and addressed all the comments in the notes below.

Please note that our responses to the reviewer’s comments are shown in red font, and line numbers mentioned here are shown in the manuscript.

Point 1: In my opinion, it would be worthwhile to consider studying one more additional group of camels with average milk yield for comparison of values, so as to make sure that the results obtained in the study were not random.

Response 1: Thank you very much for pointing out this important issue. We agree with your opinion. Unfortunately, due to the limited time, camel smples and experimental conditions, we did not supplement camels with average milk yield. This study is a continuation of the results of previous research.

  1. Transcriptome analysis of the Bactrian camel(Camelus bactrianus)reveals candidate genes affecting milk production traits, https:/doi.org/10.1186/s12864-023-09703-9).
  2. Transcriptome analysis to identify candidate genes related to mammary gland development of Bactrian camel (Camelus bactrianus), https:/doi.org/10.3389/fvets.2023.1196950.

In the research of other dairy animals, such as cattle and sheep, high and low yields are usually chosen for comparative analysis.

https://doi.org/10.1186/s12864-016-2901-1

https://doi.org/10.1016/j.gene.2022.147143

https://doi.org/10.1371/journal.pone.0220629

We understand your point that it could further strengthen our findings. However, we would like to emphasize that even without this camels with average milk yield, our manuscript provides a comprehensive and conclusive discussion on the research question. We fully understand that your suggestion is to ensure the rigor and reliability of the research. In future research, we will consider conducting high-throughput sequencing analysis on a group of camels with average milk production, such as whole genome bisulfite sequencing(WGBS).

Method

Point 2: The methodology lacks very important information regarding sampling - when how many times with what intervals, whether always at the same time, in the morning/evening if only in the morning then whether this did not affect the results obtained, how it related to the stage of lactation, whether the milk was preserved, how milk yield was determined and by what methods the basic composition of the milk was determined.

Response 2: Thanks for your professional comments. We numbered each camel and used the weighing method to track and measure the milk production of the camels. Milk yield was weighed after each milking. (Lines 449-461)

Camel milk were collected three times per month, at the beginning, middle, and end of each month. The morning and evening milk samples were collected, and immediately mixed and transported to the laboratory in an ice bath. Milk samples from each camel were not pooled together and were instead detected individually.

Samples were analyzed for solids-not-fat, fat, protein, and lactose using a fully automated milk composition analyzer (UL40BC, Youchuang).

stage of lactation

Our team has tested the milk composition during lactation in other groups of camels, so this part of the content is not our focus, so it is not shown in the manuscript. This part of the data is still being accumulated and has not been published.

Dynamic changes of camel milk composition during lactation (a) dynamic changes of protein content during lactation; (b) dynamic changes of lactose content during lactation; (c) dynamic changes of SNF content during lactation; (d) dynamic changes of ash content during lactation; (e) dynamic changes of fat content during lactation.

References

Point 3: In the literature list, the second item has an incorrect journal name.

Response 3: Thanks for reviewer’s reminder. We have conducted inspections and modifications.  Int Dary J →Int Dairy J

Round 2

Reviewer 1 Report

Comments and Suggestions for Authors

Line 114: “indicating that the nutritional composition” to be the chemical composition.

Line 125: “yielding camels (Figure 2A), the high-yielding camels possessed” to be yielding she-camels (Figure 2A), the high-yielding she-camels possessed

Point 12: Line 446: At which week she-camels reached the lactation peak?

Response 12: The peak lactation period of the Junggar Bactrian camel is from early July to the end of September each year. This is our study on milk production data during lactation weeks.

I did refer to at which week of the lactation period the she-camels reach their maximum milk yield.

Line 454: “from each camel” to be from each she-camel

Line 544: “The results of this study suggested that the nutrient content of camel” to be The results of this study suggested that the chemical composition of camel ….

Response 23: “However, our statistical analysis process and data are correct. We have checked and modified the mathematical model by adding fixed effect parameters for age and parity.”

So, what are the effects of incorporating age and parity in the model on milk yield and composition?

Comments on the Quality of English Language

Good writing style

Author Response

Response to Reviewer Comments

Line 114: “indicating that the nutritional composition” to be the chemical composition.

Response: Thanks for your comments. nutritional→chemical. Line 114

Line 125: “yielding camels (Figure 2A), the high-yielding camels possessed” to be yielding she-camels (Figure 2A), the high-yielding she-camels possessed.

Response: yielding camels→ yielding she-camels. Line 125

Line 446: At which week she-camels reached the lactation peak?

Response 12: The peak lactation period of the Junggar Bactrian camel is from early July to the end of September each year. This is our study on milk production data during lactation weeks.

I did refer to at which week of the lactation period the she-camels reach their maximum milk yield.

Response: Approximately in the fifth week after lactation and milking, the highest milk production occurs. Because the reply letter cannot insert image data, the result cannot be displayed. The data are shown in the table below.

week

Average milk yield(kg)

1

27.38

2

27.51

3

30.05

4

35.51

5

39.18

6

36.35

7

34.35

8

34.99

9

28.97

10

31.85

11

35.48

12

34.43

13

34.15

14

31.98

15

30.85

16

25.29

17

20.43

18

21.95

19

20.06

20

17.47

21

17.06

22

21.95

23

20.06

24

14.75

25

13.17

26

10.03

27

9.69

28

9.34

29

9.11

30

9.31

31

9.36

32

10.27

33

10.03

34

10.47

35

10.49

36

10.19

37

9.96

38

10.10

39

9.74

40

9.41

41

9.48

42

8.38

43

6.16

Line 454: “from each camel” to be from each she-camel

Response: camel→she-camel. Line 453

Line 544: “The results of this study suggested that the nutrient content of camel” to be The results of this study suggested that the chemical composition of camel ….

Response: nutrient content→ chemical composition. Line 543

Response 23: “However, our statistical analysis process and data are correct. We have checked and modified the mathematical model by adding fixed effect parameters for age and parity.” 

So, what are the effects of incorporating age and parity in the model on milk yield and composition?

Response: Female camels give birth to their first calf when they are about 4 or 5 years old. The inclusion of age and parturition as fixed effects parameters in the model had a significant effect on milk production and composition. To ensure the accuracy of data, We considered the effect of litter size and age factors on milk production traits. (Lines 431, 534)

Milk Yield

Age: Generally, camels tend to produce more milk as they age up to a certain point.  At the same time in the first parity, the average milk production of a 5-year-old lactating camel is higher than that of a 4-year-old lactating camel. Due to the better physical development conditions of a 5-year-old lactating camel.

Parity: Parity has a well-established effect on milk yield. First-parity cows typically produce less milk than older cows, as their bodies are still adapting to lactation.

Considering age factors in the model can more accurately predict and explain the differences in milk production among cows of different age groups.

Considering age and parity in the model can more accurately evaluate the milk production performance of lactating camels at different stages.

Milk Composition

Age and Parity: Both age and parity can affect milk composition, particularly the fat and protein content. Young camels tend to produce milk with lower fat and protein percentages, which gradually increase with age and parity. This is due to changes in the camel's metabolism and udder development over time.

Considering age and parity in the model can more accurately predict and explain the differences in the composition of camel milk produced by lactating camels under different physiological conditions.

Reviewer 2 Report

Comments and Suggestions for Authors

Dear Authors,

thank you very much for your responses to my review, they are fully sufficient for me. 

I congratulate you on your excellent work and wish you good luck with your next manuscripts.

Author Response

感谢您的意见。